# A Comprehensive Analysis of Microflora and Metabolites in the Development of Ulcerative Colitis into Colorectal Cancer Based on the Lung–Gut Correlation Theory

**DOI:** 10.3390/molecules27185838

**Published:** 2022-09-08

**Authors:** Qi Tang, Ran Liu, Ge Chu, Yue Wang, Haiyue Cui, Tongrui Zhang, Kaishun Bi, Peng Gao, Zonghua Song, Qing Li

**Affiliations:** 1School of Pharmacy, Shenyang Pharmaceutical University, 103 Wenhua Road, Shenyang 110016, China; 2School of Food and Drug, Shenzhen Polytechnic, 7098 Lau sin Avenue, Shenzhen 518000, China; 3Metabolomics Core Faculty, Feinberg School of Medicine, Northwestern University, Chicago, IL 60611, USA; 4Chinese Pharmacopoeia Commission, Building 11, Fahuananli, Beijing 100061, China

**Keywords:** ulcerative colitis, colorectal cancer, lung–gut axis, microbiota, metabolites

## Abstract

The lungs and large intestine can co-regulate inflammation and immunity through the lung–gut axis, in which the transportation of the gut microbiota and metabolites is the most important communication channel. In our previous study, not only did the composition of the gut microbiota and metabolites related to inflammation change significantly during the transition from ulcerative colitis (UC) to colorectal cancer (CRC), but the lung tissues also showed corresponding inflammatory changes, which indicated that gastrointestinal diseases can lead to pulmonary diseases. In order to elucidate the mechanisms of this lung–gut axis, metabolites in bronchoalveolar lavage fluid (BALF) and lung tissues were detected using UHPLC–Q-TOF-MS/MS technology, while microbiome characterization was performed in BALF using 16S rDNA sequencing. The levels of pulmonary metabolites changed greatly during the development of UC to CRC. Among these changes, the concentrations of linoleic acid and 7-hydroxy-3-oxocholic acid gradually increased during the development of UC to CRC. In addition, the composition of the pulmonary microbiota also changed significantly, with an increase in the *Proteobacteria* and an obvious decrease in the *Firmicutes*. These changes were consistent with our previous studies of the gut. Collectively, the microbiota and metabolites identified above might be the key markers related to lung and gut diseases, which can be used as an indication of the transition of diseases from the gut to the lung and provide a scientific basis for clinical treatment.

## 1. Introduction

Ulcerative colitis (UC) is a common non-specific inflammatory bowel disease (IBD). Long-standing extensive UC represents the main risk factor for colorectal cancer (CRC) related to IBD [1,2]. CRC is the third most commonly diagnosed cancer worldwide and the second leading cause of carcinogenic death, which threatens the health of an increasing number of individuals [3]. According to the theory of traditional Chinese medicine (TCM), the lung and gut have similar physiological functions and pathological phenomena due to their shared structural origin and connection to one another through meridians and collaterals [4,5]. As mentioned in Huangdi’s Canon of Internal Medicine, the lung and gut complement each other, forming an inseparable dependence [6,7]. Modern research has shown that patients with chronic obstructive pulmonary disease (COPD) are three times more likely to develop Crohn’s disease than healthy people, which results in gastrointestinal and respiratory tract diseases often occurring together [8]. To date, many scholars have recognized the complex interaction between the lungs and gut. A growing number of extensive studies have been conducted on the common inflammatory pathway and common immune mechanism between the respiratory system and digestive system along the lung–gut axis [9,10]. Hence, it is of paramount importance to understand the mechanism of the lung–gut axis for exploring the relationship between the lung and gut in order to gain insights into the prevention and treatment of intestinal and pulmonary diseases.

The research thus far suggests that the important communication channels of the lung–gut axis may be mediated by the circulation and transportation of the gut microbiota and metabolites, the direct migration of immune cells, the spillover of inflammatory mediators, and so on [11,12]. Among these factors, the pathways of the gut microbiota and metabolites have been confirmed to play particularly important roles in intestinal and pulmonary diseases [13]. On one hand, microbial disorder within the gut can significantly alter the levels of small molecule metabolites with immunomodulatory properties, leading to improper inflammatory responses and a series of bowel diseases [14]. Moreover, the disordered gut microbiota and metabolites can further affect the immune response in the lungs through the lung–gut axis in order to regulate the inflammation in the lungs [15,16]. Murine studies have suggested that products derived from gut microbiota, such as short-chain fatty acids (SCFAs), can modulate systemic immunity and local lung inflammation by priming the immune effector cells [17]. On the other hand, microbial communities of the respiratory and gastrointestinal tracts may also be exchanged with one another through the liquid in the lymphangion, thus directly affecting the health of the lungs [18]. Studies have found divergent compositions of gut microbiota between patients with pulmonary tuberculosis and healthy volunteers. Moreover, microbial homeostasis can be maintained through fecal transplantation, thus improving pulmonary diseases [19,20]. This phenomenon indicates that there is a complex interaction between the gut microbiota and lungs, which can affect their balance. Consequently, exploring the changing patterns in the microbiota profiles and metabolites during the development of intestinal and pulmonary diseases is of great importance for research on the interaction mechanism of the lungs and gut and the prevention and treatment of diseases. However, there are few studies on the pulmonary microbiota and their metabolites in the development of enteropathy. In addition, there are no scientific or detailed explanations determining whether intestinal diseases can cause changes in the pulmonary microbiota and their metabolites, or whether there are synchronously changed substances in the gut and lungs.

In our previous study [21], using plasma and fecal samples as a model, the changes in the small molecule metabolites from UC to CRC were measured through ultra-high-performance liquid chromatography–quadrupole time-of-flight mass spectrometry (UHPLC–Q-TOF-MS/MS). Moreover, the gut microbial communities of fecal samples were analyzed by 16S rDNA sequencing technology, which revealed different compositions and diversity in the control, UC, and CRC groups. It is noteworthy that the lung tissues also showed corresponding inflammatory changes, which provided an insight into the possibility that disturbances of the gut microbiota and metabolites caused by intestinal diseases might be transmitted to the lungs along the lung–gut axis, leading to pulmonary diseases. However, the synchronous and dynamic change patterns in the intestinal and pulmonary microbiota and metabolites during the development of diseases are still unclear. Therefore, in this study, lung tissues and BALF continued to be analyzed using UHPLC–Q-TOF-MS/MS and 16S rDNA sequencing technology, aiming to evaluate the alterations in the pulmonary metabolites and pulmonary microbiota during the development of UC into CRC. Combined with the previous experimental results, the excavated microflora and their metabolites, with the same change trend in the intestinal and pulmonary tracts, were taken as the key biomarkers related to the lungs and gut, which could be used to indicate the occurrence and development of diseases so as to provide a basis for the clinical application of TCM theory.

## 2. Materials and Methods

### 2.1. Chemicals and Reagents

2,4,6-trinitrobenzenesulfonic acid (5% *w*/*v*) and 1,2-dimethylhydrazine hydrochloride (98% purity) of chemical grade were purchased from Sigma-Aldrich (St. Louis, MO, USA). Chloral hydrate of chemical grade was purchased from Yuwang Co. Ltd. (Yucheng, China). Physiological saline was provided by Ruijinte Chemical Co. Ltd. (Tianjin, China). Ethanol of analytical grade was obtained from Hualu Pharmaceutical Co. Ltd. ((Yucheng, China). Gentiopicroside (98% purity) and dehydrocholic acid (98% purity) were purchased from Priifa Technology Development Co. Ltd. (Chengdu, China) and Yuanye Biotechnology Co. Ltd. (Shanghai, China), respectively. Methanol, acetonitrile, and formic acid of HPLC grade were provided by Fisher Scientific (Fair Lawn, New Jersey, USA). Distilled water was purchased from Wahaha Group Co. Ltd. (Hangzhou, China).

### 2.2. Animal and Modeling Process

A total of 24 Sprague-Dawley rats (aged 6 weeks, male, weighing 180–220 g) were provided by the Shenyang Pharmaceutical University (SYPU) Experimental Animal Center (Shenyang, China). All animals were raised in a specialized pathogen-free (SPF) standard chamber under a regular temperature of 22 ± 2 °C and 12 h light–dark cycle (lights on from 7 a.m. to 7 p.m.), with access to food and water ad libitum for 7 days. Protocols were verified by the SYPU Ethics Committee and carried out in line with the SYPU Guidelines for Animal Experimentation (SYPU-IACUC-2019-0509-202).

Consistent with our previous study, all rats were randomly divided into 2 groups: the control group (n = 8) and model group (n = 16, enema with 2,4,6-trinitrobenzenesulfonic acid (100 mg/kg)-50% ethanol). For the purpose of assessing the modeling success, the general conditions of the rats were observed daily, including body weight, diet and water consumption, hair color, and fecal characteristics. Moreover, several pathological parameters were observed, including the disease activity index (DAI), colonic mucosal injury index (CMDI), and hematoxylin and eosin (H&E) staining (see Appendix A and Appendix A). The successful modeling rats were stochastically classified into 2 groups: the UC group (n = 8) and CRC group (n = 8, intraperitoneal injection with 30 mg/kg 1,2-dimethylhydrazine hydrochloride once a week for 15 weeks). The CMDI, H&E staining, and tumor formation were observed to assess the success of the CRC modeling (see Appendix A). All rats were sacrificed after anesthesia to harvest bio-samples of the lungs and BALF.

### 2.3. Sample Collection and Preparation

The rats were sacrificed by cervical dislocation. After the tracheas were stripped out, the lungs were slowly flushed with 5 mL PBS buffer three times to collect the BALF. A total of 400 μL of each BALF solution was extracted using 1.2 mL methanol spiked with 10 μL internal standard solution (IS, a mixture of gentiopicroside and dehydrocholic at a final concentration of 5 μg/mL in internal standard solution). The mixture was vortexed for 3 min and centrifuged for 10 min (12000 rpm at 4 °C). The supernatant was transferred into new EP tubes and evaporated to dryness under nitrogen at 37°C. Subsequently, the extraction was reconstituted in 100 μL methanol, vortexed for 3 min, sonicated for 3 min, and then centrifuged for 5 min (12,000 rpm at 4 °C). Finally, the supernatant was obtained for subsequent analysis using the UHPLC–QTOF-MS/MS system (an Agilent 1260 HPLC system coupled with an AB SCIEX TripleTOF™ 5600 quadrupole-time-of-flight hybrid mass spectrometer system).

After the BALF samples were collected, the lung tissues were immediately harvested, followed by their washing with physiological saline and drying with filter paper. A total of 100 mg of each lung tissue was weighed and homogenized with tenfold physiological saline for 1 min. After being centrifuged (4000 rpm at 4 °C) for 10 min, the supernatant was obtained as the lung tissue homogenate. A total of 500 μL of each lung tissue homogenate was spiked with 10 μL IS and 1.5 mL methanol, vortexed for 3 min, and centrifuged (12,000 rpm, 4 °C) for 10 min to obtain the supernatant, and then dried at 37 °C in N_2_. The follow-up operation for the residue was the same as above.

The quality control (QC) samples were prepared by mixing equal aliquots of each sample. One blank sample and one QC sample were inserted for every 8 samples during the UHPLC–Q-TOF-MS/MS analysis.

### 2.4. Nontargeted Metabolomics by UHPLC–Q-TOF-MS/MS

Nontargeted metabolomics of the BALF and lungs were carried out using the Agilent 1260 Infinity HPLC system (Agilent, Santa Clara, CA, USA), coupled with a hybrid triple TripleTOF^®^ 5600+ mass spectrometer equipped with a DuoSpray™ ion source (Sciex, Foster City, CA, USA). A mobile phase consisting of 0.1% formic acid water (a) and 0.1% formic acid acetonitrile (b) at a flow rate of 0.3 mL/min was carried out on a ZORBAX SB-Aq column (100 × 2.1 mm, 1.8 μm). The gradient elution program was as follows: 2% B → 45%, 0.00 to 1.00 min; 45% B → 95% B, 1.00 to 14.00 min; 95%B, 14.00 to 20.00 min; 2% B, 20.00 to 20.01 min; and 2% B, 20.01 to 26.00 min. An m/z 50-1500 mass scope was employed as the scanning range and the detailed parameters were optimized as follows: ion spray voltage, 5500 V/−4500 V; source temperature, 550 °C; gas 1, 50 psi; gas 2, 50 psi; curtain gas, 30 psi; declustering potential, 100 V/−100 V; and collision energy, 10 V/−10 V. PeakView software (version 1.2.1, SCIEX), MarkerView software (version 1.2.1, SCIEX), and the SIMCA-P program (version14.0, Umetrics) were applied for the multivariate data analysis and processing. Metabolites identified were confirmed by comparing their information (accurate mass, MS/MS fragments) with the Human Metabolome Database (HMDB; http://www.hmdb.ca/, accessed on 2 December 2020). Pathway, enrichment, and correlation analyses were performed on MetaboAnalyst (http://www.metabo-analyst.ca/, accessed on 20 February 2021).

### 2.5. Microbial DNA Extraction and Sequencing

The genomic DNA of the samples was extracted by the CTAB method and diluted to 1ng/μL with sterile water. Then, using the diluted genomic DNA as a template, the V4 region of the bacterial 16S ribosomal gene was amplified by PCR using the 515F (GTGYCAGCMGCCGCGGTAA) and 806R (GGACTACNVGGGTWTCTAA) universal primer sequence, Phusion^®^ High-Fidelity PCR Master Mix with GC Buffer from New England Biolabs, and a high-fidelity enzyme. The TruSeq^®^ DNA PCR-Free Sample Preparation Kit library construction kit was used to construct the library. Finally, the constructed library was quantitated by Qubit and Q-PCR, and the Illumina NovaseQ6000 sequencing platform was used for the double-end sequencing.

### 2.6. Statistical Analysis of Data

QIIME^TM^ software (http://qiime.org, version 1.9.1, accessed on 3 August 2020) and R software (version 2.15.3) were used to reduce and cluster the original data in order to construct the PCA diagram, and then the Unifrac distance was calculated to construct the UPGMA sample cluster tree. In order to identify different microflora at the phylum, class, order, family and genus levels, Metastats software (http://metastats.cbcb.umd.edu/, accessed on 11 August 2020) was used for the ANOVA of the microflora in different groups. In addition, the correlation between the biomarkers of the UC and CRC groups identified in this study and those previously found in the plasma and feces was analyzed. Ultimately, a Pearson correlation analysis of the pulmonary microbiome and metabolites was performed using SPSS software, and the correlation matrix was drawn.

## 3. Results

### 3.1. Metabolomics Data Analysis

Before the analysis of the formal samples, the precision, reproducibility, and stability of the established method were evaluated. Six ions with a superior peak type and higher peak intensity were selected from the QC samples. Moreover, the relative standard deviation (RSD) of the peak intensity and retention time were used as the methodology validation indexes. The results showed that the RSD of the retention time ranged from 0.1% to 4.0%, and the RSD of the peak strength ranged from 4.2% to 14.9%, indicating that the established method of analysis was suitable for our analysis (see Appendix A).

The original data were processed using MarkerView^TM^ software (version 1.3.1, SCIEX) to obtain the peak identification, peak alignment, and normalization, and the missing values were filtered in accordance with the 80% principle using Excel software. Then, the pretreated data were imported to SIMCA-P software (version14.1, Umetrics, Umea, Sweden), and the principal components analysis (PCA) of the lung tissue and BALF samples was conducted. Samples from the control, UC, and CRC groups were highly differentiated, as represented by green, blue, and red dots, respectively. Then, orthogonal least squares discriminant analysis (OPLS-DA) models were established to further reflect the differences between the groups and identify differential metabolites. As shown in Figure 1, both the UC and CRC groups were highly different from control. The values of the R2 and Q2 were greater than 0.99 and 0.5, respectively, indicating the good fitting degree and prediction ability of the OPLS-DA models. In addition, no over-fitting phenomena were found in the permutation tests, proving that the OPLS-DA models were reliable. The results above indicated that the pulmonary metabolic activities were significantly affected in the development of UC into CRC.

After the analysis of the PCA and OPLS-DA, the affected metabolites were screened according to the following criteria: VIP values greater than 1 in the OPLS-DA, p values less than 0.05 in the t-test, and fold change values greater than 2 or less than 0.5. The metabolite identities were confirmed by comparing the information of MS and MS2 with the HMDB database (http://www.hmdb.ca/, accessed on 2 December 2020). Ultimately, 19 differential metabolites were found in both the BALF and lung tissues of the UC group. A total of 12 differential metabolites were found in the BALF and 21 biomarkers were found from the lung tissue of the CRC group. Among these, 15 differential metabolites were detected in both the UC and CRC groups, including 3-methylxanthine, tetrahydrocortisone, N-acetylaminooctanoic acid, (9E,11E)-octadecadienoic acid, 12b-hydroxy-5b-cholanoic acid, S-lactoylglutathione, 7-hydroxy-3-oxocholanoic acid, deoxyinosine, oleamide, phytosphingosine, LysoPE(18:3(6Z,9Z,12Z)/0:0), LysoPE(20:4(5Z,8Z,11Z,14Z)/0:0), LysoPC(18:2(9Z,12Z)/0:0), L-arginine, and linoleic acid. In particular, 7-hydroxy-3-oxocholanoic acid and linoleic acid were also found in the plasma and fecal samples. Their changing trends were consistent with our previous results, which indicated that these two metabolites can be used as key biomarkers related to the lungs and large intestine. The molecular formula, molecular weight, change trend, and other information about the biomarkers are shown in Table 1.

MetaboAnalyst 4.0 (http://www.metaboanalyst.ca/, accessed on 20 February 2021) was used to conduct a hierarchical cluster analysis of the three groups (Figure 2A). The concentrations in the different groups varied significantly. The metabolic pathway analysis (Figure 2B) using MetaboAnalyst 4.0 revealed 15 metabolic pathways which may play important roles in UC, and 13 metabolic pathways in the occurrence of CRC. According to the comprehensive analysis, nine pathways may be closely related to the development of UC into CRC, including linoleic acid metabolism, arginine biosynthesis, sphingolipid metabolism, pyruvate metabolism, the biosynthesis of unsaturated fatty acids, glycerophospholipid metabolism, arginine and proline metabolism, and aminoacyl-tRNA biosynthesis, as well as purine metabolism. Compared with our previous findings, linoleic acid metabolism, sphingolipid metabolism, and the biosynthesis of unsaturated fatty acids were also identified in the results of the plasma and fecal samples. Therefore, these three pathways showed potential as key metabolic pathways related to the lungs and gut.

### 3.2. Intestinal Microflora Analysis

The original data obtained using the Illumina Novaseq platform were filtered through Reads Mosaic to obtain the effective data. An OTU (operational taxonomic unit) cluster analysis was also performed. Then, an alpha diversity calculation was carried out to obtain information on the species richness and evenness across samples. As shown in Appendix A, the coverage represented the probability that the bacterial community in the sample was detected, and these values were all greater than 0.997, thus proving that our sequencing results were reliable. Based on this finding, the rarefaction curve and the rank abundance curve (Appendix A) were generated to obtain the information about the species diversity, richness, and evenness. Compared with the control group, the species diversity, richness, and evenness of the CRC group decreased markedly, while, on the contrary, these parameters in the UC group were slightly increased.

Subsequently, QIIME^TM^ software was used to quantitatively analyze the microbiota in the BALF at the levels of the phylum, class, order, family, and genus. The histograms of the species distribution were drawn with the R language tool, and the top 10 abundant species at different levels are shown in Figure 3A. The altered compositions of the pulmonary microbiota in the three groups revealed the disorder of the microbial community during the intestinal disease. Interestingly, at the phylum level, *Firmicutes* and *Actinobacteria* were enriched in the three groups, whereas at the class level, *Bacilli* was enriched in the control, while *Gammaproteobacteria* was enriched in both the UC and CRC groups. These results were consistent with our previous study of gut microbiota, which indicated that the microbiota in the intestinal and pulmonary tracts are similar to some extent.

Moreover, QIIME^TM^ software was used to calculate the Unifrac distance and construct the UPGMA sample cluster tree. A PCA diagram (Figure 3B,C) was drawn using R software. From the PCA results, we can see that the control group was not significantly in contrast with the UC group but was clearly distinguished from the CRC group. The UPGMA results also showed a significantly greater distance between the control and CRC groups, while the distance between the control and UC groups was not especially noteworthy. It is suggested that the risk of pulmonary disease in CRC group was higher than that in UC group. Finally, the ANOVA analysis was used with Metastats software to select the differential flora (P values of less than 0.05) in a more accurate manner (Table 2). The results showed that, compared with the control group, the differential microflora of the UC group were *Firmicutes*, *Proteobacteria* (at the phylum level) and *Bacilli* (at the class level), and the differential microflora of the CRC group were *Firmicutes* and *Proteobacteria* (at the phylum level); *Bacilli*, *Clostridia*, *Alphaproteobacteria*, and *Gammaproteobacteria* (at the class level); *Corynebacteriales*, *Clostridiales*, *Rhizobiales*, and *Pasteurellales* (at the order level); *Corynebacteriaceae*, *Lachnospiraceae*, *Ruminococcaceae*, *Rhizobiaceae*, *Burkholderiaceae*, *Xanthobacteraceae*, and *Pasteurellaceae* (at the family level); and *Staphylococcus*, *Shinella*, *Bradyrhizobium*, and *Rodentibacter* (at the genus level). Among these microbial communities, compared to the control group, the abundance of *Firmicutes* was decreased in the UC group and even lower in the CRC group. On the contrary, the abundance of *Proteobacteria* was increased in the UC group and even higher in the CRC group. These alterations were consistent with the results of our previous gut microbial community study, which further verified the scientific basis of the “lung–gut correlation” theories.

### 3.3. Correlation between the Metabolite and Pulmonary–Intestinal Microecology

In order to verify the interaction between the differential metabolites in the gut and lungs, the correlation between the differential metabolites found in this study and the differential metabolites found in the plasma and feces was analyzed (Figure 4A). The results showed that most of the correlation coefficients of the differential metabolites in the plasma, feces, BALF, and lung tissue were greater than 0.8, indicating that the intestinal metabolism and pulmonary metabolism can interact with each other.

Furthermore, explicit correlation coefficients were calculated in SPSS (version 20.0, IBM Corp., Armonk, NY, USA), using the Pearson method to investigate the correlation of the metabolite pulmonary–intestinal microecology (Figure 4B). The correlation coefficient of 15 differential metabolites that showed effects on both the UC and CRC and differential microflora were visualized by a matrix graph. About 3.2% of the correlation coefficient values were between 0.5 and 0.6, 7.0% were between 0.6 and 0.7, 28.7% were between 0.7 and 0.8, and 57.7% were in excess of 0.8. Moreover, 7-hydroxy-3-oxocholanoic acid and linoleic acid showed significant positive correlations with *Proteobacteria* (the correlation coefficients were 0.896 and 0.986, respectively) and significant negative correlations with *Firmicutes* (the correlation coefficients were −0.79 and −0.876, respectively), which were consistent with the results of the intestinal microflora and their metabolites. In conclusion, the gut and lungs can affect each other’s balance, and gut dysbiosis can alter the level of metabolites and influence lung homeostasis via the lung–gut axis, thus promoting the occurrence and development of diseases.

## 4. Discussion

In present study, high-throughput 16S rDNA gene sequencing and metabolomic analysis were used to investigate the effects of the progression of UC to CRC on the pulmonary microbiota and their metabolic profiles. The data clearly showed that the intestinal disease altered the composition of the pulmonary microbiota and their metabolites. In addition, the pulmonary microbiome was associated with a large number of pulmonary metabolites, suggesting that the progression of UC to CRC not only disturbed the balance of the pulmonary microflora, but also substantially altered their metabolomic profiles. Moreover, the correlation analysis of the differential metabolites in the plasma, feces, BALF, and lung tissue showed that most of the correlation coefficients were greater than 0.8, indicating that the intestinal metabolism and pulmonary metabolism can affect each other. These findings provide mechanistic insights into the “lung–gut correlation” theory.

From the results of metabolomic analyses, 15 metabolites with great variations in both the UC and CRC groups were identified. Among them, linoleic acid and 7-hydroxy-3-oxocholanoic acid were also found in our previous research and showed the same change trend. Additionally, metabolic pathways of the 15 differential metabolites were analyzed, and 9 metabolic pathways were eventually identified. Linoleic acid metabolism, sphingolipid metabolism, and the biosynthesis of unsaturated fatty acids were also identified in our previous study. In both studies, linoleic acid metabolism was the most influential. As a member of the ω-6 polyunsaturated fatty acid group, linoleic acid is known to increase the levels of cytokines, which leads to neutrophilia [22]. Individuals with more linoleic acid in their diet have an increased risk of UC [23]. Moreover, human diets with vegetable oil rich in linoleic acid can increase the proliferation of cancer cells in the colon [24]. Several scholars have noted that linoleic acid also plays an important role in pulmonary diseases. For example, linoleic acid can enhance severe asthma by causing airway epithelial damage. In addition, non-small-cell lung cancer is usually characterized by the mutation of the epidermal growth factor receptor (EGFR), and the level of linoleic acid tends to increase significantly in cancer patients with more EGFR [25,26]. Therefore, linoleic acid is highly correlated with intestinal and pulmonary diseases and could be a key biomarker for monitoring the occurrence of pulmonary and intestinal diseases.

L-arginine was involved in three pathways, including aminoacyl-tRNA biosynthesis, arginine biosynthesis, and arginine and proline metabolism. Arginine is reported to have a significant effect on immune regulation, promoting intestinal development and anti-tumor and anti-obesity effects [27,28]. As a precursor of NO synthesis, arginine inhibits the expression of matrix metalloproteinases and cell adhesion molecules, thereby preventing cell adhesion and inhibiting tumor cell proliferation [29,30]. In the results of the metabolomics, compared with the control group, the content of L-arginine was decreased in the UC group and was even lower in the CRC group, which suggests that the decreasing trend of L-arginine may be influenced by the intestinal tumor cells.

Arginine can be synthesized into purine in vivo by means of arginase. Purine plays a pivotal role in energy supply, metabolism regulation, coenzyme composition, and the inflammatory immune response, and it is also closely related to the occurrence of pain [31,32]. In purine metabolism, we detected a decreased content of 3-methylxanthine in the UC group and an even lower content in the CRC group. Methylxanthine was involved in numerous regulatory roles, such as increasing the blood circulation, dilating the blood vessels, improving the airflow, reducing inflammation, and preventing chronic obstructive pulmonary disease, which is highly applicable to the treatment of respiratory diseases [33,34]. Thus, the reduced concentration of 3-methylxanthine could be associated with pulmonary disease caused by intestinal diseases.

In addition, in pyruvate metabolism, the concentration of S-lactoylglutathione in the control, UC and CRC groups showed a gradual downward trend. Excessive free radicals produced by organism metabolism damage the biofilm and induce the generation of tumors. The main physiological functions of glutathione include anti-free radical, anti-aging, and anti-oxidation functions [35,36]. Thus, glutathione can eliminate free radicals and show a powerful protective effect in enhancing human immunity [37,38]. Therefore, the reduced concentration of S-lactoylglutathione also reflected the inflammatory response in the lungs caused by intestinal diseases.

The 16S rDNA sequencing results showed that the compositions of the lung microflora of the three groups varied greatly. The imbalance of the pulmonary microbiota caused by gut dysbiosis demonstrated the correlation between the lungs and gut. By means of the *t*-test, a number of differential microflora were screened (*p* < 0.05), including *Firmicutes*, *Proteobacteria*, and *Bacilli* (between the control and the UC groups), and *Firmicutes*, *Proteobacteria*, *Clostridia*, *Alphaproteobacteria*, *Bacilli*, *Gammaproteobacteria*, *Corynebacteriales*, *Clostridiales*, *Rhizobiales*, *Pasteurellales*, *Corynebacteriaceae*, *Lachnospiraceae*, *Ruminococcaceae*, *Rhizobiaceae*, *Xanthobacteraceae*, *Pasteurellaceae*, *Staphylococcus*, *Shinella*, *Bradyrhizobium*, and *Rodentibacter* (between the control and CRC groups). These microbial communities can be classified into three groups: *Actinobacteria*, *Firmicutes*, and *Proteobacteria*. Among them, the abundant of *Firmicutes* and *Proteobacteria* gradually decreased and gradually increased during the development of UC to CRC, respectively, findings which were consistent with what we observed in the gut microbiota. Furthermore, there are many studies that have found results similar to ours. Charlson et al. observed a significant increase in *Proteobacteria* in COPD rats compared with the control group. Moreover, exposure to cigarette smoke can dysregulate the gut microbiota and further aggravate inflammation in the lungs. In an even more novel finding, it was observed that the intestines of rats from the COPD group exhibited mucosa of a darkened gray color and significant swelling [39]. Another study also found that TH2-low asthma was associated with the increase in *Proteobacteria* and the decrease in *Firmicutes* by comparing the compositions of the pulmonary microbiota among different asthma endotypes [17]. These results fully proved that the lung and gut are interrelated in the development of diseases, and the increase in the abundance of *Proteobacteria* may lead to intestinal and pulmonary diseases.

Furthermore, the results of the gut microbiota study showed that the abundance of *Gammaproteobacteria*, *Enterobacteriales*, *Enterobacteriaceae*, and *Escherichia-Shigella* also gradually increased during the development of the disease. However, in the results of the pulmonary microbiota study, the same trend was only found in the *Gammaproteobacteria*, while *Enterobacteriales*, *Enterobacteriaceae*, or *Escherichia-Shigella* were not detected. The reason for this might be that *Enterobacteriales*, *Enterobacteriaceae*, and *Escherichia-Shigella* were mostly present in the gut. Moreover, compared with the control group, the *Bacilli* in the intestinal tract were in lower abundance in the UC group and even lower abundance in the CRC group. In the pulmonary flora study, although *Bacilli* was a differential flora in the both UC and CRC groups, the abundance of *Bacilli* did not show a gradual decreasing trend during the disease; thus, *Bacilli* was not regarded as the key microflora related to the lungs and gut.

Apart from these results, no changes in the *Rhizobiales*, *Rhizobiaceae*, *Shinella*, *Xanthobacteraceae*, *Bradyrhizobium*, and other microflora were detected in the gut microbiota. However, the significant changes in the structural composition of the pulmonary microflora and the correlation results indicated that the gut microbiota can affect the pulmonary microbiota and their metabolic activities. In recent years, a great number of studies have demonstrated the bidirectional nature of the lung–gut axis and that the intestinal and pulmonary microbiota can influence each other. The overall influence of the healthy airway microbiota on host immunity reflects the cumulative effects of microbes and their metabolites on local and systemic innate and adaptive immune processes in the host [40]. If the process of healthy and timely colonization is disrupted, early-life dysbiosis of the intestinal and pulmonary microbiota becomes an important risk factor for the development of many respiratory diseases [41]. All these studies demonstrated the scientific basis of our conclusions and confirm the interaction between the lungs and gut.

## 5. Conclusions

Based on the “lung–gut correlation” theory, the structure of the intestinal microbiota and untargeted metabolomics were analyzed, and the results were combined with our previous research data to evaluate the microbiota and metabolites in the gut and lungs during the development of UC into CRC. From the metabolomic results, linoleic acid and 7-hydroxy-3-oxocholanoic acid were discovered to be the key metabolites related to lung and gut diseases. The pulmonary microbiota study results showed that *Firmicutes* and *Proteobacteria* may be the key microflora associated with the lung and gut diseases, and the decrease in *Firmicutes* or the increase in *Proteobacteria* may cause intestinal and pulmonary diseases. Finally, the correlation of the metabolite pulmonary–intestinal microecology was proved using a Pearson correlation analysis. Our findings support the notion that intestinal diseases can lead to pulmonary diseases by affecting the changes in the microbiota and metabolites. In conclusion, our study identified the key biomarkers related to lung and intestine, which are critical for signaling the transition from gastrointestinal diseases to pulmonary diseases. This finding is an important contribution to the development and innovation of the TCM theory.

## Figures and Tables

**Figure 1 molecules-27-05838-f001:**
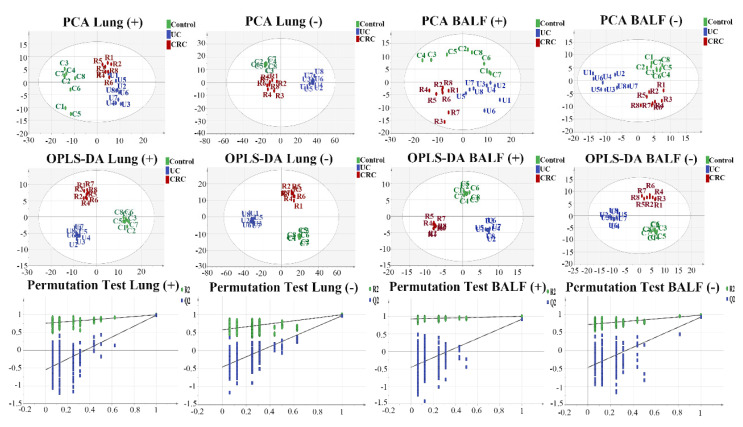
The PCA score plots of the control, UC, and CRC groups for the BALF in ESI (+), R2 = 0.562; BALF in ESI (−), R2 = 0.572; for the lung tissue in ESI (+), R2 = 0.577; lung tissue in ESI (−), R2 = 0.732. Green, blue, and red dots are used to label the control, UC, and CRC groups, respectively. OPLS-DA score plots of the control, UC, and CRC groups for the BALF in ESI (+), R2 = 0.995, Q2 = 0.879; BALF in ESI (−), R2 = 0.982, Q2 = 0.909; lung tissue in ESI (+), R2 = 0.987, Q2 = 0.915; lung tissue in ESI (−), R2 = 0.993, Q2 = 0.965. Green, blue, and red dots are used to label the control, UC, and CRC groups, respectively. Validation plots of the OPLS-DA models obtained using the 200 permutation tests for the BALF in ESI (+), R2 = 0.922, Q2 = −0.457; BALF in ESI (−), R2 = 0.715, Q2 = −0.504; lung tissue in ESI (+), R2 = 0.761, Q2 = −0.480; lung tissue in ESI (−), R2 = 0.574, Q2 = −0.405. Green and blue dots stand for the R2 and Q2, respectively.

**Figure 2 molecules-27-05838-f002:**
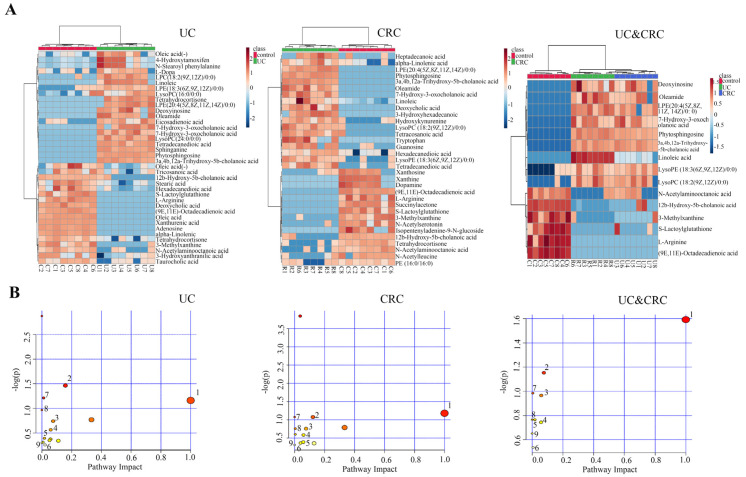
(**A**) The hierarchical clustering heatmap. (**B**) Pathway analysis overview showing altered metabolic pathways in the UC group: (1) linoleic acid metabolism, (2) sphingolipid metabolism, (3) purine metabolism, (4) biosynthesis of unsaturated fatty acids, (5) taurine and hypotaurine metabolism, (6) alpha-linolenic acid metabolism, (7) arginine biosynthesis, (8) pyruvate metabolism, (9) glycerophospholipid metabolism, (10) arginine and proline metabolism, (11) tryptophan metabolism, (12) tyrosine metabolism, (13) primary bile acid biosynthesis, (14) aminoacv-tRNA biosvnthesis, (15) drug metabolism—cytochrome P450. In the CRC group: (1) linoleic acid metabolism, (2) purine metabolism, (3) biosynthesis of unsaturated fatty acids, (4) glycerophospholipid metabolism, (5) alpha-linolenic acid metabolism, (6) glycosylphosphatidylinositol (GPI)-anchor biosynthesis, (7) arginine biosynthesis, (8) sphingolipid metabolism, (9) pyruvate metabolism, (10) arginine and proline metabolism, (11) tryptophan metabolism, (12) tyrosine metabolism, (13) aminoacyl-tRNA biosynthesis. In the UC and CRC groups: (1) linoleic acid metabolism, (2) arginine biosynthesis, (3) sphingolipid metabolism, (4) pyruvate metabolism, (5) biosynthesis of unsaturated fatty acids, (6) glycerophospholipid metabolism, (7) arginine and proline metabolism, (8) aminoacyl-tRNA biosynthesis, (9) purine metabolism.

**Figure 3 molecules-27-05838-f003:**
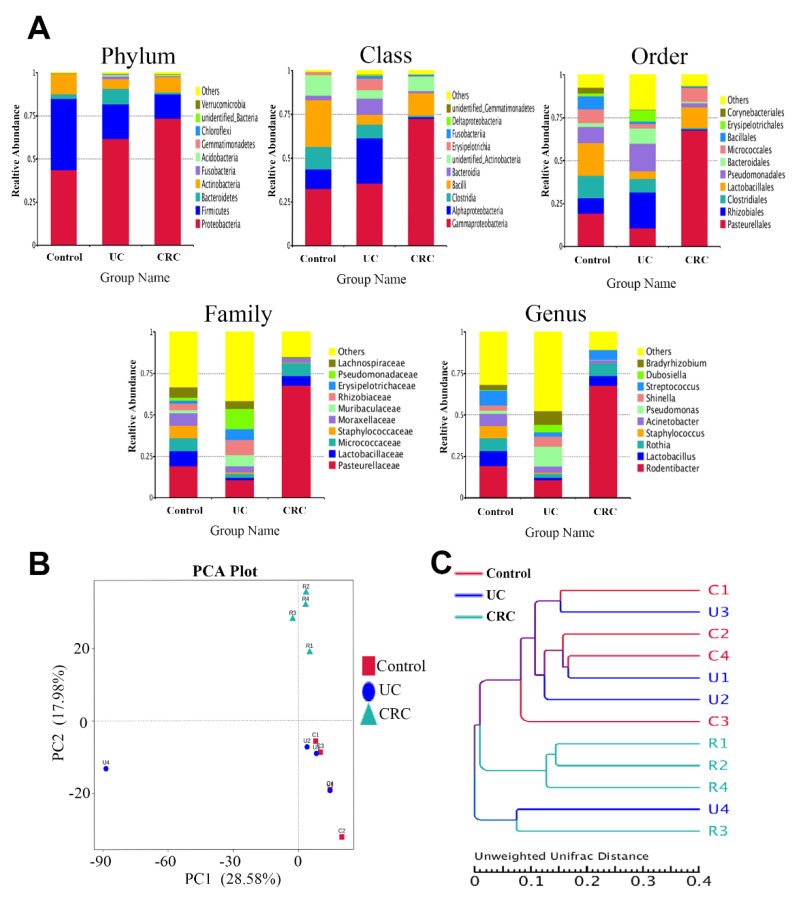
(**A**) Relative abundance of pulmonary microbiota at the phylum, class, order, family, and genus levels in the control, UC, and CRC groups. (**B**) PCA analysis diagram. (**C**) Sample UPGMA cluster tree.

**Figure 4 molecules-27-05838-f004:**
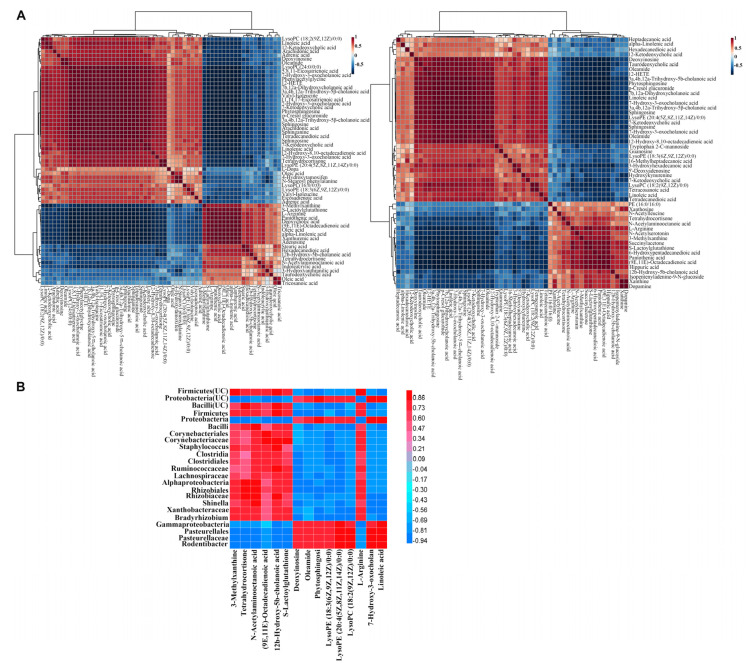
(**A**) Analysis of the biomarker−to−biomarker correlations in different samples. (**B**) Correlation plot showing the functional correlation between the pulmonary microflora and metabolites.

**Table 1 molecules-27-05838-t001:** The detailed information about the potential biomarkers detected in the ulcerative colitis (UC) group and colorectal cancer (CRC) group.

No.	t(R)	ESI Mode	*m*/*z*	Identification	Fomular	VIP1	VIP2	Fold1	Fold2	Source	Change Trend		
											UC/Con.	CRC/Con.	CRC/UC
1	3.1	+	153.1354	3-Hydroxyanthranilic acid	C_7_H_7_NO_3_	1.29		3.55		BALF UC	↓		
2	3.3	+	166.1374	3-Methylxanthine	C_6_H_6_N_4_O_2_	1.69	1.74	7.02	12.95	BALF UC&CRC	↓	↓	↓
3	9.5	+	197.1879	L-Dopa	C_8_H_15_N_5_O	1.49		0.43		BALF UC	↑		
4	6.3	+	282.4614	Oleic acid	C_18_H_34_O_2_	1.25		0.16		BALF UC	↑		
5	9.1	+	364.4758	Tetrahydrocortisone	C_21_H_32_O_5_	1.30	1.71	3.66	6.35	BALF UC&CRC	↓	↓	↓
6	3.1	+	365.3900	Isopentenyladenine-9-N-glucoside	C_16_H_23_N_5_O_5_		1.84		68.03	BALF CRC		↓	
7	6.3	+	387.5140	4-Hydroxytamoxifen	C_26_H_29_NO_2_	1.25		0.08		BALF UC	↑		
8	6.4	+	432.2784	N-Stearoyl phenylalanine	C_27_H_45_NO_3_	1.22		0.06		BALF UC	↑		
9	11.5	+	495.6301	LysoPC(16:0/0:0)	C_24_H_50_NO_7_P	1.02		0.45		BALF UC	↑		
10	6.4	+	515.7030	Taurocholic acid	C_26_H_45_NO_7_S	1.60		3.85		BALF UC	↓		
11	12.1	−	201.2628	N-Acetylaminooctanoic acid	C_10_H_19_NO_3_	1.13	1.69	8.30	5.09	BALF UC&CRC	↓	↓	↓
12	3.0	−	218.2518	N-Acetylserotonin	C_12_H_14_N_2_O_2_		1.55		8.30	BALF CRC		↑	
13	11.3	−	258.3538	Tetradecanedioic acid	C_14_H_26_O_4_		1.37		0.19	BALF CRC		↑	
14	10.8	−	278.4296	alpha-Linolenic acid	C_18_H_30_O_2_		1.04		0.31	BALF CRC		↑	
15	14.6	−	280.4455	(9E,11E)-Octadecadienoic acid	C_18_H_32_O_2_	1.80	3.23	4.30	13428.75	BALF UC&CRC	↓	↓	↓
16	6.6	−	283.2407	Guanosine	C_10_H_13_N_5_O_5_		1.16		0.23	BALF CRC		↑	
17	15.4	−	282.4614	Oleic acid	C_18_H_34_O_2_	2.39		0.01		BALF UC	↑		
18	16.5	−	284.4772	Stearic acid	C_18_H_36_O_2_	2.67		0.01		BALF UC	↑		
19	16.5	−	286.4070	Hexadecanedioic acid	C_16_H_30_O_4_	1.86		10.59		BALF UC	↓		
20	16.0	−	308.4986	Eicosadienoic acid	C_20_H_36_O_2_	1.76		0.01		BALF UC	↑		
21	19.1	−	354.6101	Tricosanoic acid	C_23_H_46_O_2_	1.34		5.07		BALF UC	↓		
22	19.6	−	368.6367	Tetracosanoic acid	C_24_H_48_O_2_		2.10		0.17	BALF CRC		↑	
23	14.3	−	376.5726	12b-Hydroxy-5b-cholanoic acid	C_24_H_40_O_3_	2.24	2.31	0.01	0.01	BALF UC&CRC	↑	↑	↑
24	3.1	−	379.3860	S-Lactoylglutathione	C_13_H_21_N_3_O_8_S	1.33	2.36	6.47	192.22	BALF UC&CRC	↓	↓	↓
25	9.2	−	390.5561	7-Hydroxy-3-oxocholanoic acid	C_24_H_38_O_4_	1.65		0.01		BALF UC	↑		
26	2.3	+	158.1519	Succinylacetone	C_7_H_10_O_4_		1.63		101.76	Lung CRC		↓	
27	2.1	+	174.2010	L-Arginine	C_6_H_14_N_4_O_2_	1.18	1.19	10.08	15.72	Lung UC&CRC	↓	↓	↓
28	4.8	+	224.2133	Hydroxykynurenine	C_10_H_12_N_2_O_4_		1.07		0.23	Lung CRC		↑	
29	2.3	+	252.2300	Deoxyinosine	C_10_H_12_N_4_O_4_	1.07	1.33	0.04	0.04	Lung UC&CRC	↑	↑	↑
30	8.0	+	281.2719	Oleamide	C_18_H_35_NO	1.38	2.14	0.01	0.01	Lung UC&CRC	↑	↑	↑
31	12.6	+	301.5078	Sphinganine	C_18_H_39_NO_2_	1.93		0.01		Lung UC	↑		
32	11.8	+	317.5072	Phytosphingosine	C_18_H_39_NO_3_	2.04	2.34	0.01	0.01	Lung UC&CRC	↑	↑	↑
33	7.8	+	364.4758	Tetrahydrocortisone	C_21_H_32_O_5_	1.25		0.03		Lung UC	↑		
34	10.9	+	366.3700	Tryptophan 2-C-mannoside	C_17_H_22_N_2_O_7_		1.21		0.02	Lung CRC		↑	
35	5.3	+	475.5558	LysoPE (18:3(6Z,9Z,12Z)/0:0)	C_23_H_42_NO_7_P	1.35	1.44	0.03	0.14	Lung UC&CRC	↑	↑	↑
36	9.6	+	501.2855	LysoPE (20:4(5Z,8Z,11Z,14Z)/0:0)	C_25_H_44_NO_7_P	1.52	1.93	0.01	0.01	Lung UC&CRC	↑	↑	↑
37	5.2	+	519.3325	LysoPC (18:2(9Z,12Z)/0:0)	C_26_H_50_NO_7_P	1.39	1.74	0.05	0.04	Lung UC&CRC	↑	↑	↑
38	5.3	+	607.4577	LysoPC(24:0/0:0)	C_32_H_66_NO_7_P	1.57		0.01		Lung UC	↑		
39	2.4	−	152.1109	Xanthine	C_5_H_4_N_4_O_2_		1.91		456.69	Lung CRC		↓	
40	2.4	−	153.1784	Dopamine	C_8_H_11_NO_2_		1.33		17.24	Lung CRC		↓	
41	9.6	−	173.2096	N-Acetylleucine	C_8_H_15_NO_3_		1.27		4.90	Lung CRC		↓	
42	4.8	−	205.2099	Xanthurenic acid	C_10_H_7_NO_4_	1.55		1376.79		Lung UC	↓		
43	7.8	−	258.3538	Tetradecanedioic acid	C_14_H_26_O_4_	1.40		0.01		Lung UC	↑		
44	7.0	−	267.2413	Adenosine	C_10_H_13_N_5_O_4_	1.42		452.62		Lung UC	↓		
45	12.0	−	270.4507	Heptadecanoic acid	C_17_H_34_O_2_		1.28		0.15	Lung CRC		↑	
46	11.6	−	272.4290	3-Hydroxyhexadecanoic acid	C_16_H_32_O_3_		1.93		0.01	Lung CRC		↑	
47	10.3	−	278.4296	alpha-Linolenic acid	C_18_H_30_O_2_	1.57		1722.66		Lung UC	↓		
48	13.4	−	280.4455	Linoleic acid	C_18_H_32_O_2_	1.33	1.76	0.05	0.04	Lung UC&CRC	↑	↑	
49	5.0	−	282.4614	Oleic acid	C_18_H_34_O_2_	1.52		1017.46		Lung UC	↓		
50	3.7	−	284.2255	Xanthosine	C_10_H_12_N_4_O_6_		1.03		6.26	Lung CRC		↓	
51	9.7	−	286.4070	Hexadecanedioic acid	C_16_H_30_O_4_		1.10		0.28	Lung CRC		↑	
52	8.1	−	390.2770	7-Hydroxy-3-oxocholanoic acid	C_24_H_38_O_4_	1.43	1.61	0.01	0.12	Lung UC&CRC	↑	↑	↑
53	9.7	−	392.5720	Deoxycholic acid	C_24_H_40_O_4_	1.47		773.12		Lung UC	↓		
54	7.5	−	408.5714	3a,4b,12a-Trihydroxy-5b-cholanoic acid	C_24_H_40_O_5_	1.68	1.70	0.01	0.01	Lung UC&CRC	↑	↑	↑
55	20.2	−	691.9720	PE (16:0/16:0)	C_37_H_74_NO_8_P		1.07		0.01	Lung CRC		↑	

UC: ulcerative colitis group, CRC: colorectal cancer group, Con.: control group. VIP1: control group/UC group, VIP2: control group/CRC group. Fold1: control group/UC group, Fold 2: control group/CRC group, ↑: increased, ↓: decreased.

**Table 2 molecules-27-05838-t002:** Different pulmonary microorganisms between the ulcerative colitis (UC) group and the colorectal cancer (CRC) group at the levels of the phylum, class, order, family, and genus.

	Ulcerative Colitis	Colorectal Cancer
Phylum	Firmicutes, Proteobacteria	Firmicutes, Proteobacteria
Class	Bacilli	Bacilli, Clostridia, Alphaproteobacteria
		Gammaproteobacteria
Order	/	Corynebacteriales, Clostridiales, Rhizobiales,
		Pasteurellales
Family	/	Corynebacteriaceae, Lachnospiraceae, Ruminococcaceae, Rhizobiaceae
		Xanthobacteraceae, Pasteurellaceae,
Genus	/	Staphylococcus, Shinella, Bradyrhizobium, Rodentibacter

## Data Availability

Not applicable.

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
