# Peer review of "A Comprehensive Analysis of Microflora and Metabolites in the Development of Ulcerative Colitis into Colorectal Cancer Based on the Lung–Gut Correlation Theory"

_molecules, 2022, doi:10.3390/molecules27185838_

Round 1

Reviewer 1 Report

The authors of the presented manuscript continue their previous work on investigating the lung-gut correlation theory through pulmonary microbiota analysis and lung tissue/fluid metabolomics assessment during the transition from ulcerative colitis (UC) to colorectal cancer (CRC). The authors provide comprehensive data elucidating the key pro-inflammatory markers and pulmonary microbiota being in relation to lung and gut diseases in a way that ensured the metabolites-pulmonary-intestinal-microecology axis. The manuscript is well-structured, and the work can be considered as valuable addition to the fields of pathophysiology and clinical management.

Publication of this review manuscript is recommended following minor suggestions:

1.    In animal and modeling process section; the age of the used rats should be mentioned since organ development timeline for gut (mucosal immunity) generally extends beyond the birth, the thing that make such organ more vulnerable towards detrimental injuries. Thus, reporting the rat age is important to assess such possibility.   

2.  The authors reported the use of t-test statistical analysis. However, in section 3.2 lines 303-305, the authors suggested different abundance of Firmicutes and Proteobacteria in CRC and UC groups in relation to controls as well as to each other. Thus, authors duplicated the use of t-test across the three groups (control, CRC, and UC) for same organism data (Firmicutes or Proteobacteria). Using AVOVA is more appropriate here to compare the means of more than two population groups, equating them together, and avoid Type I errors that would result from conducting multiple t-tests on the same data.

3.    In lines 221-223, could the authors rationalize the adopted screening cut-offs (VIP > 1; fold change > 2 and < 0.5) and what was the evidence for adopting such values.

4.   In discussion section, the authors just rephrased their results findings comparing their major finding with only their previous study, making discussion just like another result section. I would suggest the authors to enhance their theoretical discussion in a better way through comparing their results with more relevant research work and similar studies reported within current literature, either for gut or lung in correlation to other organs or in correlation to each other.

5. In Figure S1, image resolution should be improved, as well as a comprehensive figure legend should be introduced describing the main histopathological findings within each image that would be also highlighted with colored arrows on the images themselves. Additionally, the image size scale should be notified.

6.      In Table S3, the ± SD values are missing.

Author Response

Thank you so much for reviewing and giving us the chance to revise our manuscript # molecules-1879068 entitled “Comprehensive analysis of microflora and metabolites in the development of ulcerative colitis to colorectal cancer based on the lung-gut correlation theory". We greatly appreciate the efforts for handling and reviewing our manuscript as well as the valuable comments. Please find our response to the comments below and the relative changes made in the revised manuscript.

Point 1: In animal and modeling process section; the age of the used rats should be mentioned since organ development timeline for gut (mucosal immunity) generally extends beyond the birth, the thing that make such organ more vulnerable towards detrimental injuries. Thus, reporting the rat age is important to assess such possibility.

Response 1: Thank you for your advice. It is really true as Reviewer suggested that the age of the used rats should be mentioned. The age of the used rats in our experiment was 6 weeks. We have indicated the age of the rats in the manuscript and marked in red.

Point 2: The authors reported the use of t-test statistical analysis. However, in section 3.2 lines 303-305, the authors suggested different abundance of Firmicutes and Proteobacteria in CRC and UC groups in relation to controls as well as to each other. Thus, authors duplicated the use of t-test across the three groups (control, CRC, and UC) for same organism data (Firmicutes or Proteobacteria). Using ANOVA is more appropriate here to compare the means of more than two population groups, equating them together, and avoid Type I errors that would result from conducting multiple t-tests on the same data.

Response 2: Thank you for your advice. As Reviewer suggested that ANOVA is more appropriate. We have made correction according to your comments in the revised manuscript.

Point 3: In lines 221-223, could the authors rationalize the adopted screening cut-offs (VIP > 1; fold change > 2 and < 0.5) and what was the evidence for adopting such values.

Response 3: For metabolomics, the most commonly used indicators for screening differential metabolites are as follows: 1.P value; 2. Fold Change (FC); 3. VIP value.

P value is a probability, which reflects the probability of an event and is used to distinguish whether the variable has statistical significance. It is usually considered as P<0.05 was statistically significant; Fold Change refers to the calculation of the difference in the expression of a metabolite between two groups according to the relative or absolute quantitative results of metabolites. The usual thresholds are 2 (values greater than this threshold are retained) and 0.5 (values less than the threshold are retained); The full name of VIP value is Variable Influence on Projection, which is obtained according to OPLS-DA method and reflects the degree of contribution of each expression to the model. A threshold of 1 is usually set.

Using these three indicators to screen differential metabolites is the most commonly used and very scientific and effective method in metabolomics research. We also referred to a large number of literatures, and the following are some references that use this method to screen differential metabolites: 1.UPLC-MS based urine untargeted metabolomic analyses to differentiate bladder cancer from renal cell carcinoma. 2. Protective effect of ethyl eosmarinate against ulcerative colitis in mice based on untargeted metabolomics.

Point 4: In discussion section, the authors just rephrased their results findings comparing their major finding with only their previous study, making discussion just like another result section. I would suggest the authors to enhance their theoretical discussion in a better way through comparing their results with more relevant research work and similar studies reported within current literature, either for gut or lung in correlation to other organs or in correlation to each other.

Response 4: Thank you so much for your advice. Considering your suggestion, we have added a several of relevant research work and similar studies in our manuscript and marked in red. And we found that these studies really enhance our theoretical discussion in a better way.

Point 5: In Figure S1, image resolution should be improved, as well as a comprehensive figure legend should be introduced describing the main histopathological findings within each image that would be also highlighted with colored arrows on the images themselves. Additionally, the image size scale should be notified.

Response 5: Thank you so much for your advice. We have carefully modified the image as you suggested, improving the resolution and highlighting the main pathological findings with colored arrows. In addition, the image size scale has been notified (HE, 400× magnification). The red arrows represent the loss of mucus in the goblet cells; The green arrows represent inflammatory cell infiltration; The yellow arrows represent intestinal epithelial vacuoles.

Point 6: In Table S3, the ± SD values are missing.

Response 6: Thank you so much for reviewing our manuscript. After careful examination, we found that x ± SD was a writing error, and Table S3 showed the OTU, Shannon, Simpson, Chao1, ACE and coverage values detected for each sample instead of x ± SD. And we have deleted “x ± SD” in the title of Table S3.

Reviewer 2 Report

In the manuscript, the chemical structures of microflora metabolites are missing. Please add them in a separate manuscript. 

Improve the quality of all figures. It is rarely visible the words in the figures.

Analyze the metabolomics data analysis with the help of database 

Author Response

Thank you so much for reviewing and giving us the chance to revise our manuscript # molecules-1879068 entitled “Comprehensive analysis of microflora and metabolites in the development of ulcerative colitis to colorectal cancer based on the lung-gut correlation theory". We greatly appreciate the efforts for handling and reviewing our manuscript as well as the valuable comments. Please find our response to the comments below and the relative changes made in the revised manuscript.

Point 1: In the manuscript, the chemical structures of microflora metabolites are missing. Please add them in a separate manuscript.

Response 1: Thank you so much for reviewing our manuscript. As Reviewer suggested that we have added the chemical structure of microflora metabolites in a separate manuscript and thank you for your advice.

Point 2: Improve the quality of all figures. It is rarely visible the words in the figures.

Response 2: Thank you for your advice, we have improved the quality of all figures. And all figures were at high resolution and the words in the figures have been very clearly.

Point 3: Analyze the metabolomics data analysis with the help of database.

Response 3: Thank you so much for reviewing our manuscript. Metabolites identities were confirmed by comparing information of MS and MS2 with HMDB database. I'm sorry for not writing the information clearly when we submitted the manuscript for the first time. We have added the information of HMDB database in the manuscript and marked it in red.
